# Typing to Listen at the Cocktail Party: Text-Guided Target Speaker Extraction

## Abstract

Humans possess an extraordinary ability to selectively focus on the sound source of interest amidst complex acoustic environments, commonly referred to as cocktail party scenarios. In an attempt to replicate this remarkable auditory attention capability in machines, target speaker extraction (TSE) models have been developed. These models leverage the pre-registered cues of the target speaker to extract the sound source of interest. However, the effectiveness of these models is hindered in real-world scenarios due to the unreliable or even absence of pre-registered cues. To address this limitation, this study investigates the integration of natural language description to enhance the feasibility, controllability, and performance of existing TSE models. Specifically, we propose a model named LLM-TSE, wherein a large language model (LLM) extracts useful semantic cues from the user's typed text input. These cues can serve as independent extraction cues, task selectors to control the TSE process or complement the pre-registered cues. Our experimental results demonstrate competitive performance when only text-based cues are presented, the effectiveness of using input text as a task selector, and a new state-of-the-art when combining text-based cues with pre-registered cues. To our knowledge, this is the first study to successfully incorporate LLMs to guide target speaker extraction, which can be a cornerstone for cocktail party problem research. Demos are provided at https://github.com/LLM-TSE/llm-tse.github.io[1]

## 1 Introduction

The "Cocktail Party Problem" (E. Colin, 1953) – a term coined to describe a scenario where multiple sound sources are engaged in simultaneous conversation, yet a listener can selectively concentrate on a single sound source. This scenario represents a complex challenge in auditory perception (Haykin & Chen, 2005; Mesgarani & Chang, 2012; Bizley & Cohen, 2013) and serves as a remarkable demonstration of the intricate sound processing that occurs within the human auditory system. The human auditory system manages this complexity with remarkable efficacy, seemingly with ease. However, machines, such as hearing-aid devices (Shinn-Cunningham & Best, 2008), teleconferencing systems (Chen et al., 2020; Raj et al., 2021; Yoshioka et al., 2018), and hands-free human-machine interfaces (e.g., TVs, smartphones) (Gannot et al., 2017), encounter significant challenges in the context where multiple speakers talk at the same time.

Studies on computational auditory scene analysis (CASA) (Lyon, 1983; Meddis & Hewitt, 1991; Seltzer et al., 2003; Wang & Brown, 2006), non-negative matrix factorization (NMF) (Cichocki et al., 2006; Virtanen, 2007; Parry & Essa, 2007), and factorial Hidden Markov Models and Gaussian Mixture Models (HMM-GMM) (Virtanen, 2006; Stark et al., 2011) provide invaluable insights into solving the cocktail party problem. However, these methods are often limited by the representation power of their models, resulting in poor performance in complex acoustic environments. The advent of deep learning has paved the way for the application of deep neural networks (DNNs) in addressing this challenging problem. These existing DNN-based techniques can be broadly classified into two main categories: blind source separation (BSS) (Pal et al., 2013; Hershey et al., 2016; Yu et al., 2017; Luo & Mesgarani, 2019) and target speaker extraction (TSE) (Luo et al., 2018; Žmolíková et al., 2019; Xu et al., 2020; Ge et al., 2020; Pan et al., 2022; Zmolikova et al., 2023).

---

[1]Source code and datasets will be publicly available after review.

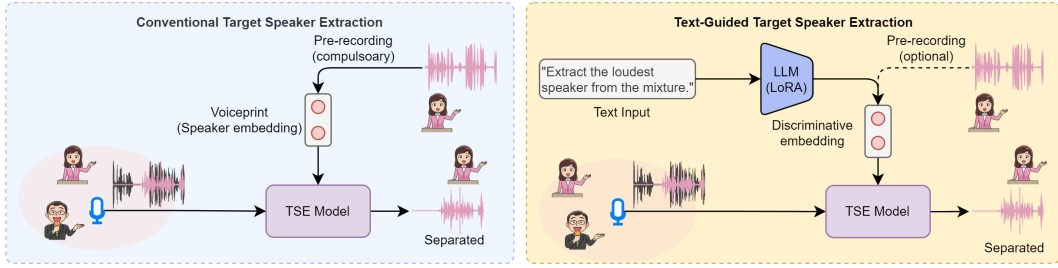

Figure 1: Comparison between the conventional TSE system and our proposed Text-Guided TSE system. The conventional systems rely on the pre-registered voiceprint of the target speaker as an extraction cue, while our system offers the flexibility to incorporate text-based cues to facilitate the target speaker extraction.

BSS techniques usually adopts DNNs to estimate an auditory mask for each speaker. The mask is then leveraged to separate each speaker's voice into an individual stream from the mixture speech captured by a microphone. A difficulty in this process is the problem of global permutation ambiguity (Hershey et al., 2016), which occurs when attempting to accurately assign the output of a multi-source separation system to the correct source. To address this ambiguity problem, deep clustering (DC) techniques (Hershey et al., 2016; Isik et al., 2016; Wang et al., 2018) were proposed to group the spectro-temporal features belonging to the same speaker together through a clustering scheme. Permutation invariant training (PIT) (Yu et al., 2017; Kolbæk et al., 2017) was invented by finding the minimal loss over all the permutations between the extracted streams and the reference speeches. Typically, these methods require prior knowledge or estimation of the number of speakers in the mixture. However, in real-world scenarios, the number of speakers is hard to predict in advance.

Target speaker extraction provides an alternative solution to address the challenges of the unknown number of speakers and global permutation ambiguity. This approach involves providing a cue that is related to the desired speaker, such as a pre-recorded speech describing the voice characteristics (Xu et al., 2020), a spatial cue indicating the speaker's direction (Ge et al., 2022), or synchronous lip movement (Pan et al., 2022). By using these specified cues, only the target speaker's voice is extracted, thereby avoiding the issue of the unknown number of speakers and global permutation ambiguity. However, these pre-registered cues may vary substantially or even be absent in real environments, limiting the effectiveness of these systems.

To overcome the aforementioned limitation, as shown in Figure 1, we propose a novel text-guided TSE model, LLM-TSE, incorporating text descriptions as additional cues to enhance the feasibility, controllability, and performance of existing TSE models. Specifically, we leverage the power of large language models (LLMs) to extract meaningful semantic cues from the user's typed text input. These text descriptions encompass various aspects of human auditory perception, including speaker characteristics, language, conversation contents, room characteristics, etc. These cues can serve as independent extraction cues, task selectors to control the TSE process or complement the pre-registered cues. By incorporating text descriptions as additional cues, we demonstrate that the performance of TSE models is significantly enhanced in various scenarios. The contributions of this work can be summarised as follows:

- To the best of our knowledge, this is the first study to utilize natural language description as extraction cues for target speaker extraction. We show these semantic cues possess high discriminative power and, therefore, can significantly enhance the feasibility of existing TSE methods.

- Our system implements a control mechanism through the natural language description to facilitate the speaker extraction process. This approach enables us to selectively retain or remove the source of interest based on the semantic concepts expressed in the text. By using text as a control mechanism, our system becomes a unified and flexible approach that eliminates the need for training multiple systems.

- Our system represents a significant advancement in TSE by integrating context-dependent information from typed descriptions with pre-registered cues. Unlike traditional cues, typically pre-recorded and isolated from the current acoustic environment, Our system captures complement cues from human perception. By incorporating additional cues that align with human

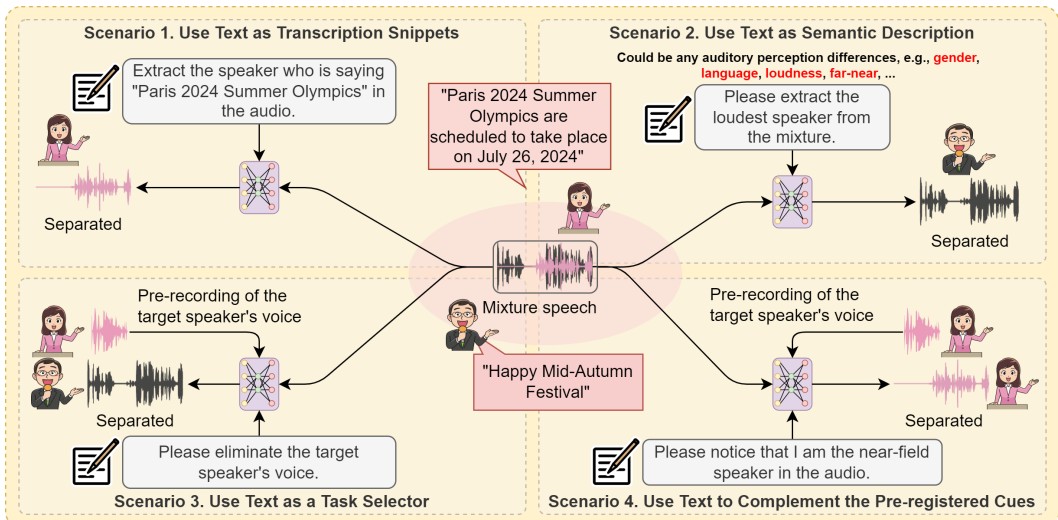

Figure 2: New application scenarios enabled by the proposed LLM-TSE model.

perception, our system achieves a more accurate and comprehensive representation of speech mixtures, thereby improving the effectiveness of TSE in practical scenarios.

## 2 TEXT-GUIDED TARGET SPEAKER EXTRACTION

The proposed LLM-TSE model opens up a plethora of novel application scenarios, surpassing the capabilities of traditional TSE techniques. As depicted in Figure 2, these application scenarios can be divided into the following four categories:

**Use text as transcription snippets:** Humans utilize discernible cues in relatively clean speech segments to enhance the perception of highly corrupted speech segments. Analogously, the LLM-TSE model can leverage distinguishable acoustic cues, in the form of transcription snippets, to facilitate speaker extraction, surpassing the capabilities of current TSE models.

**Use text as semantic description:** Apart from the above content-based cues, humans employ many other perceptual cues based on the distinguishing characteristics between competing speakers, such as gender, language, loudness level, and reverberation in the audio signal. The LLM-TSE model enables users to incorporate such perceptual cues as text-based semantic descriptions to exert control over the process of target speaker extraction. Notably, these perceptual cues can be considered as independent pre-registered cues.

**Use text as a task selector:** During a conversation involving multiple speakers, humans often switch their focus from one speaker to another. In addition, the speaker of interest at one moment may become a distraction at a later moment. In contrast to existing TSE systems that can only concentrate on a pre-registered speaker, the proposed LLM-TSE model empowers users with the flexibility to decide whether to retain or exclude the pre-registered speaker from the audio mixture, expanding the capabilities beyond what is currently achievable.

**Use text to complement the pre-registered cues.** In conventional TSE systems, the voice of the target speaker is typically pre-recorded in an acoustic environment that may differ substantially from the actual deployment environments. This discrepancy significantly affects the robustness of conventional TSE systems. In contrast, the proposed LLM-TTS model has the ability to compensate for these differences by providing complementary cues in addition to the pre-registered ones, such as the speaker's location, language, loudness level, etc. Consequently, it generates a more comprehensive and accurate representation of the target speaker that can significantly enhance the system's robustness.

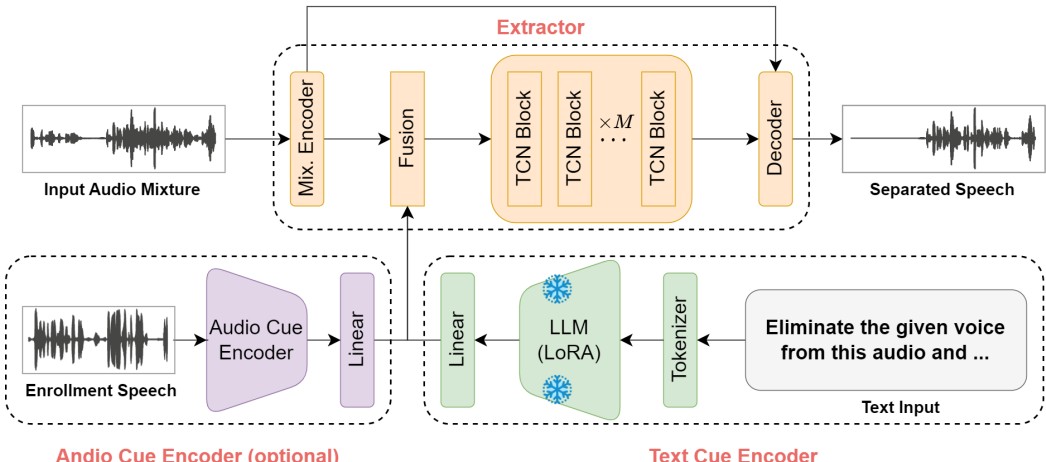

Figure 3: Overview of the proposed LLM-TSE model architecture.

## 3 LLM-TSE MODEL

As illustrated in Figure 3, the proposed LLM-TSE model follows a processing pipeline of Encoding-Fusion-Extraction-Decoding. In the encoding phase, three distinct encoders are employed to convert the pre-registered speech, text prompts, and input audio mixture into corresponding embeddings. Leveraging the fused embeddings representing the enrolled speech and text cues, the extractor then selectively extracts the desired sound source from the input audio mixture. Finally, the frequency-domain feature representation obtained from the extractor is transformed back into the time-domain and output as the extracted speech.

**Mixture Encoder and Decoder:** The mixture encoder transforms the input audio mixture from the time domain into the feature representation, which can be more effectively handled by the extractor. This transformation is realized by convolving each audio frame of length $L$ with a set of $N$ 1-D convolution filters $\{u_n(t)\}_{n=\{0...N-1\}}$, which can be expressed as follows:

$$\mathbf{X}(k, n) = \sum_{t=0}^{L-1} x(t + kH)u_n(t), \quad n \in \{0, \ldots, N-1\}, \tag{1}$$

where $x(t)$ is the input signal, $k \in \{0, \ldots, K-1\}$ is the frame index, $H$ is the hop size, and $\mathbf{X}(k, n)$ is the result of the convolution operation. Similarly, the decoder maps the extracted feature, denoted as $\mathbf{Y}(k, n)$, back to the time domain via a transposed 1-D convolution operation with $N$ synthesis filters $\{v_n(t)\}_{n=\{0...N-1\}}$, and each has a length of $L$:

$$\hat{y}(t) = \sum_{k=0}^{K-1} \sum_{n=0}^{N-1} \mathbf{Y}(k, n)v_n(t - kH), \tag{2}$$

where $\hat{y}(t)$ is the extracted audio signal in time domain.

**Text Cue Encoder:** We utilize the LLaMA-2 7B Chat LLM, a dialogue-fine-tuned version of the LLaMA-2 (Touvron et al., 2023), to obtain discriminative semantic embeddings from the user's text input. LLaMA-2 is pre-trained on a combination of natural language and programming language corpora in a self-supervised manner. LLaMA-2 7B Chat LLM is further fine-tuned from LLaMA-2 via instruction-tuning, which significantly enhances its performance on various reasoning and generation tasks. During our model training, instead of performing full fine-tuning on the adopted LLM text encoder, we adopt the parameter-efficient Low-Rank Adaptation (LoRA) technique (Hu et al., 2021). LoRA introduces a small set of parameters into the frozen LLaMA-2 7B Chat LLM, which are referred to as LoRA adapters. Specifically, one LoRA adapter is attached to each LLM layer, modifying its frozen parameter by adding a low-rank learnable matrix of the same size. In the proposed LLM-TSE model, we apply the LoRA adapters to only modify keys and queries in each self-attention layer. Ultimately, we only add 12% more trainable parameters. This approach

not only helps to prevent the overfitting problem that is often encountered with a small fine-tuning dataset but also improves the training efficiency.

**Audio Cue Encoder:** The primary role of the audio cue encoder is to encode the optional pre-registered speech into a discriminative speaker embedding. The first step in this process involves transforming the time domain input signal, using the above-mentioned learnable 1-D convolutional filters, into the frequency domain. Following this transformation, we utilize a series of Temporal Convolutional Network (TCN) blocks (Pandey & Wang, 2019; Luo & Mesgarani, 2019) to extract speaker-related feature representation. These TCN blocks are designed to capture the temporal dependencies in the speech signal, which are crucial for distinguishing different speakers. Finally, we take the average along the temporal dimension to generate a speaker embedding vector, which effectively captures the unique vocal attributes of the pre-registered speech that can differentiate one speaker from others.

**Fusion Layer:** Here, we follow a simple concatenation approach to fuse the audio and text cues, which has shown to be effective in many other TSE systems (Žmolíková et al., 2019; Ge et al., 2020). Specifically, we transform the text cue and audio cue embeddings into the same dimensionality through two linear projection layers, and then directly concatenate them to form a multi-modal representation.

**Extractor:** The last part of our model is the target extractor, which serves to estimate the target signal. We adopt the widely used time-frequency masking-based extractor (Luo & Mesgarani, 2019; Isik et al., 2016), whose operations can be summarized as follows:

$$
\begin{aligned}
\mathbf{M} &= \text{MaskNet}(\mathbf{Z}; \theta^{\text{Mask}}), \\
\hat{\mathbf{X}} &= \mathbf{M} \otimes \mathbf{X},
\end{aligned}
\tag{3}
$$

where $\mathbf{Z}$ is the fused embedding generated from the fusion layer, $\text{MaskNet}(\cdot)$ is a TCN-based NN that estimates the time-frequency mask $\mathbf{M} \in \mathbb{R}^{D \times N}$ for the target speaker, where $D$ is the feature dimension of each time step. $\theta^{\text{Mask}}$ is the network parameter, and $\otimes$ denotes the element-wise Hadamard product. $\hat{\mathbf{X}}$ is the estimated target speech signal in the frequency domain.

**Loss function:** The parameters of the proposed LLM-TSE model are optimized by minimizing the following Scale-Invariant Signal-to-Distortion Ratio (SI-SDR) (Roux et al., 2019) loss function:

$$
\mathcal{L}^{\text{SI-SDR}} = -10 \log_{10} \left( \frac{\| \frac{\hat{\mathbf{y}}^T \mathbf{y}}{\|\mathbf{y}\|^2} \mathbf{y} \|^2}{\| \frac{\hat{\mathbf{y}}^T \mathbf{y}}{\|\mathbf{y}\|^2} \mathbf{y} - \hat{\mathbf{y}} \|^2} \right).
\tag{4}
$$

The SI-SDR loss is computed directly in the time domain, which forces the model to learn how to precisely estimate the magnitude and the phase of the target speech signals.

## 4 EXPERIMENTAL EVALUATION

In this paper, our primary objective is to integrate text-based cues to enhance the target speaker extraction systems. In the following sections, we initially delve into the method of simulating the overlapped mixture of speech data. Subsequently, we will explore the generation of text questions.

### 4.1 OVERLAPPED SPEECH SIMULATION

Our experiment uses two speech datasets: LibriSpeech (Panayotov et al., 2015) and Multilingual LibriSpeech (MLS) (Pratap et al., 2020). LibriSpeech, a 1000-hour corpus of English audiobook speech, is known for its diverse speaker identities. MLS, an extension of LibriSpeech, adds multiple languages, including French, German, Spanish, etc. Due to it having too much data, we randomly selected 400 speakers per language from MLS with up to 20 utterances each. We adhered to LibriSpeech's standard training, validation, and test set division. For MLS, we randomly assigned 5% of speakers from each language to validation and test sets, respectively, with the rest for training.

Our experiments cover a variety of attributes, including transcription snippets, gender, language, loudness, and far-near. For transcription snippets extraction, we only use the LibriSpeech dataset

and the corresponding pre-extracted forced alignment (Chodroff, 2023) data [2] to identify the word timestamps from LibriSpeech. The remainder of the data for simulation is randomly selected from the LibriSpeech and MLS datasets. For generating the mixture speech, we adopt online simulation, generating the data needed for each iteration beforehand. The number of speakers in the mixture of speech is limited to two, stipulating that the two speakers have different attributes for gender, language, loudness, or far-near. When generating a mixture of speech for the loudness task, our signal-to-noise ratio is randomly selected from -3 dB to -2 dB and 2 dB to 3 dB. The other tasks span from -3 dB to 3 dB. In the case of the distance task, we include both near (target speaker) - far (interference speaker) and far (interference speaker) - near (target speaker) scenarios. For the other tasks, near and far combinations are randomized. Room dimensions are randomly selected from lengths of 9 to 11 m, widths of 9 to 11 m, and heights of 2.6 to 3.5 m. The reverberation time ranges from 0.3 to 0.6 seconds. We use Pyroomacoustics [3] to generate Room Impulse Responses (RIRs), and the microphone's position is defaulted to the center of the room. The sound source distance from the microphone varies between 0.3 to 0.5 m and 1.5 to 2.5 m for near or far fields, respectively. The angle ranges from 0 to 180 degree, and the sound source's height varies between 1.6 to 1.9 m.

The mixture and pre-registered speeches are set to a duration of 6 seconds, with a randomly determined overlap ratio between 40% and 70%. The pre-registered speech is randomly selected from the remaining target speaker's speech. If the training objective is to remove the target speaker, the other speaker's speech from the mixture serves as the training target. We assume that each generated mixture speech sample should exhibit a distinguishable attribute throughout the training. All experimental data is sampled at 16,000 Hz to ensure high-quality audio.

## 4.2 TEXT GENERATION

We include three types of texts to explore using LLMs to enrich target speaker extraction systems. We first create ten foundational question templates for each type of task. These templates will then be rephrased and expanded using ChatGPT-4-32K [4] to produce 100 diverse text prompts. We adopt a non-overlapped 80/10/10% partitioning for training, validation, and testing sets. The text prompts used in the testing set are unseen during the training.

**Text as an independent extraction cue:** In this type, the text is used as an independent extraction cue. The texts of this task are like: "Extracting a voice with ⟨ specific characteristic ⟩ from a mixture of speech", e.g., scenarios 1&2 in Figure 2. The text description outlines the features of the voice to be extracted, including the transcription snippets of the mixture of speech, the speaker's language, gender, loudness, and far-near. For the transcription snippet task, we used 100% of the target speech text length as cues for training, testing with 50%, 80%, and 100% of the target speech text length to evaluate generalizability. This setup is highly functional, i.e., by informing the system about the audible part of the speech, the system can utilize both semantic and acoustic information to track and extract the desired speaker. It's crucial to note that the attributes utilized in this study are not exhaustive. In real-world situations, humans employ a variety of other cues, e.g., emotion, pitch, etc., to extract the sound source of interest (Haykin & Chen, 2005; Shinn-Cunningham & Best, 2008). However, exploring these additional cues extends beyond the scope of this current study and is reserved for future research.

**Text as a task selector:** We propose one task type where texts can influence the system's output: target speaker extraction or removal. The text serves as a directive for the system to either extract a given speaker's voice or remove it from the mixture of audio. The generated texts are like "please remove the given voice from this audio."

**Text as a complement to human perception in the audio-based extraction system:** We integrate the human understanding and interpretation of the mixture of speech into the extraction process, which can significantly enhance the system's performance. Here, we cover all semantic types mentioned above, i.e., transcription snippets, gender, language, loudness, and far-near. The generated questions are like "Extracting a speaker based on the given pre-registered speech, where the speaker possesses a ⟨specific characteristic⟩ within the mixture speech."

---

[2]https://github.com/CorentinJ/librispeech-alignments
[3]https://github.com/LCAV/pyroomacoustics
[4]https://platform.openai.com/docs/models

Table 1: Evaluation of SI-SDR (dB ↑) metric across different methods. For the transcription snippet task, we use 100% of the target speech text as cues during training and test the model with a different amount of text transcriptions, including 50%, 80%, and 100%.

| Entry | Inputs | | Transcription Snippet | | | Gender | Language | Far-near | Loudness |
|---|---|---|---|---|---|---|---|---|---|
| | Audio | Text | 50% | 80% | 100% | | | | |
| Unproc. | - | | | -0.02 | | -0.02 | -0.03 | -0.01 | -0.10 |
| TD-SpeakerBeam | ✓ | ✗ | | 7.21 | | 10.15 | 8.38 | 9.38 | 7.57 |
| LLM-TSE (LoRA Adapters, LLaMA-2 7B Chat) | ✓ | ✗ | | 7.30 | | 10.17 | 8.87 | 9.77 | 7.75 |
| | ✗ | One-Hot | | No Support | | 10.54 | 8.88 | 10.25 | 8.96 |
| | ✗ | ✓ | 2.70 | 3.97 | 7.48 | 10.40 | 9.38 | 10.57 | 8.89 |
| | ✓ | One-Hot | | No Support | | 10.62 | 10.18 | 10.32 | 8.99 |
| | ✓ | ✓ | 7.96 | 9.81 | 10.05 | 10.87 | 9.72 | 10.66 | 9.41 |
| No LoRA Adapters (only Linear Projection) | ✗ | ✓ | 1.66 | 3.38 | 5.38 | 8.76 | 7.38 | 8.45 | 5.46 |
| | ✓ | ✓ | 4.85 | 7.60 | 7.98 | 9.02 | 7.97 | 8.67 | 7.11 |
| Use Vicuna-7b-v1.3 (Zheng et al. (2023)) | ✗ | ✓ | 2.23 | 3.31 | 8.79 | 9.44 | 8.29 | 9.27 | 5.75 |
| | ✓ | ✓ | 7.41 | 9.05 | 9.35 | 10.15 | 9.01 | 9.94 | 6.47 |

## 4.3 RESULTS

**Efficacy of Using Input Text as Independent Cues:** Table 1 demonstrates a notable performance enhancement when text alone is employed as an extraction cue, compared to unprocessed mixture speech. The proposed LLM-TSE model is built on TD-SpeakerBeam (Delcroix et al., 2020), a state-of-the-art (SOTA) open-source target speaker extraction model. Compared to TD-SpeakerBeam, the only modification in the LLM-TSE model is the additional text-prompt encoder. This enhancement is further corroborated by Figure 4. These findings suggest that the LLM-TSE model effectively interprets the provided text descriptions, which fundamentally serve as human interpretations of auditory object differences within a speech mixture. This innovative strategy represents a significant leap in harnessing natural language processing techniques for complex auditory tasks, thereby enhancing the scope of potential applications for speaker extraction methodologies.

**Efficacy of Using Input Text as Task Selector:** In this task, our objective is to control the training targets of the separation system using natural language. The corresponding textual queries could resemble "Is there a way to remove the given voice from this mixture audio?" In Figure 4, we illustrate the capacity of our system to determine whether to extract or suppress the sound source corresponding to the provided pre-registered speech when using text descriptions. Notably, the samples displayed in the third row exemplify this capability, as they successfully suppress the target sound source associated with the pre-registered speech. Our explorations in this area are somewhat limited at this stage. More broadly, these controls could be configured with greater flexibility. For instance, they could manipulate the degree of reverberation in the extracted speech (since individual preferences for reverberation vary) or dictate the impact range of the separation system (to avoid unnecessary non-linear-processing distortion). We intend to delve deeper into these aspects in our future work.

**Efficacy of Using Input Text to Complement the Pre-registered Cues:** Pre-registered speech primarily only encodes the speaker's vocal characteristics regardless of any time or acoustic environmental context. We aim to introduce this contextual information into the target speaker extraction system utilizing text descriptions. For this purpose, a typical text description is like: "Separate the target speaker's audio based on the provided pre-registered speech as a reference, bearing in mind that I am the speaker who employs a louder tone in the mixed speech." The relevant experimental outcomes are presented in the middle section of Table 1. Upon integrating descriptions delineating auditory object differences, we noted a significant improvement in the system's performance. This enhancement was particularly prominent in the "loudness" task, where the dataset contained a pronounced loudness disparity between the two sound sources. The challenge posed by identifying the target speaker using only the pre-registered speech was substantially mitigated upon implementing our approach, producing the most substantial performance increase within this task.

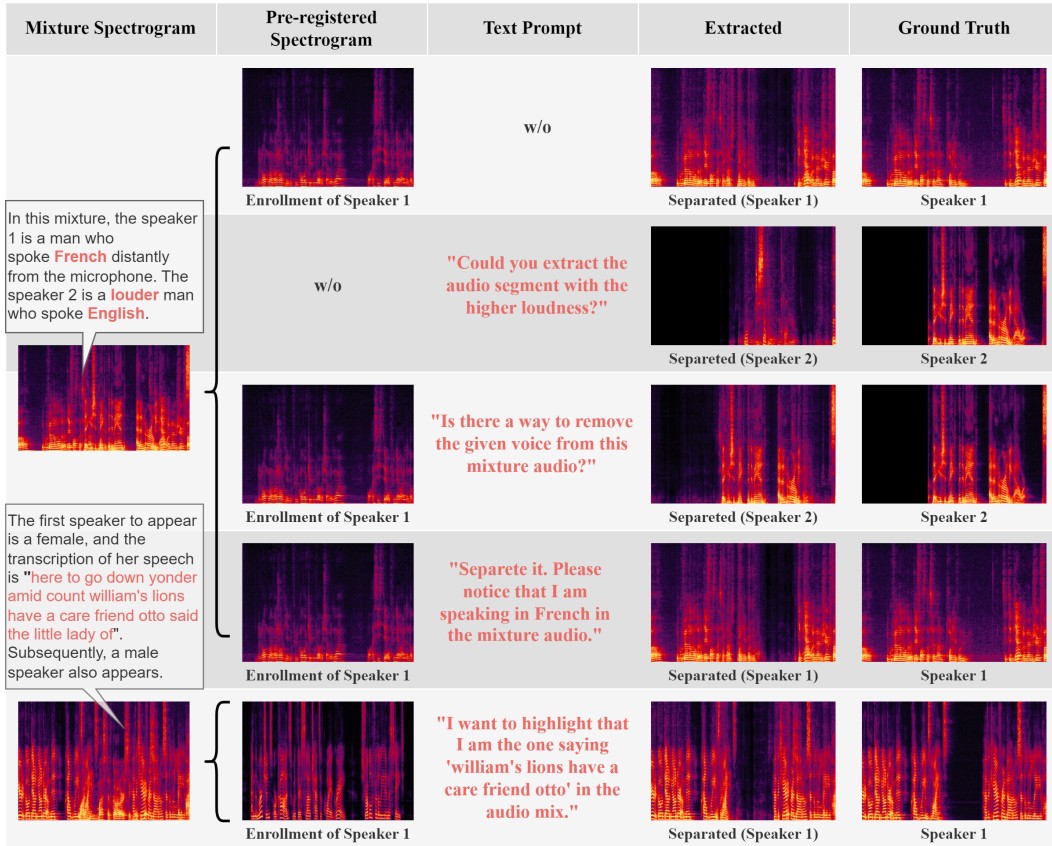

Figure 4: Samples generated from the proposed LLM-TSE model. The text box contains information about the input audio mixture. The term "w/o" indicates the absence of a certain input.

**Ablation Studies on Text Encoder Selection:** Here, we present the results of a sequence of ablation experiments executed on the text encoder component. The outcomes are summarized at the bottom of Table 1. At the outset, we assessed the functionality of the text cue encoder in the absence of the LoRA adaptors, where only the projection layer of the LLM model was permitted to train, effectively freezing all other parameters of the LLM. This configuration aimed to determine if the LLM's generic understanding of diverse text corpora could offer sufficient discriminative information. However, our findings suggest that relying solely on embeddings, which are derived from the LLM's interpretation of various text descriptions, is insufficient to accomplish the task whether an audio encoder was integrated into the system or not. In subsequent experiments, we employed the Vicuna 7B model (Zheng et al., 2023) as our text encoder. This model, which was fine-tuned on data from "shareGPT.com" and based on the LLaMA-v1 model, exhibited marginally inferior performance in natural language benchmark tasks compared to the LLAMA-2 7B Chat. Further, the Vicuna model underperformed in our target speaker separation task compared to the LLAMA-2 7B Chat. This observation supports the premise that employing a more powerful LLM as a text cue encoder can significantly enhance the discriminative capabilities of the overall system.

## 5 RELATED WORKS

**Audio-Language Multimodal Model:** Audio-language multimodal currently represents a significant research area with many application scenarios (Huang et al., 2023c; Zhang et al., 2023; Gong et al., 2023). The primary focus has revolved around audio events, with most tasks and datasets originating from automatic audio caption (Drossos et al., 2017; Wu et al., 2019; Mei et al., 2022), which aims to assign meaningful textual descriptions to audio content. Leveraging these datasets, related studies have been conducted on synthesizing audio based on text descriptions, which find applications in diverse scenarios such as film production, game design, and more. Among these, the Contrastive Language-Audio Pretraining (CLAP) (Elizalde et al., 2022) model is a large-scale

pre-training model that employs a contrastive learning approach similar to the Contrastive Language-Image Pretraining (CLIP) (Radford et al., 2021) model for aligning text and audio modalities. This model has pushed the boundaries in tasks that involve synthesizing audio based on text descriptions (Huang et al., 2023b; Kreuk et al., 2023; Liu et al., 2023a;b). Furthermore, the works conducted by Wang et al. (2023); Zhang et al. (2023); Le et al. (2023) expands the input modality to encompass audio and text instead of text only for audio generation. However, it's important to note that the underlying logic is based on generative models that take audio and specific control inputs to handle various speech transformation tasks. These works are more like controlled speech/audio/music synthesis, not requiring the length of input and output to be strictly aligned. This is entirely different from the field of our study.

**Audio-Language-Vison Multimodal Target Source Separation:** Among all these audio-language multimodal models, those most relevant to our research involve separating or detecting audio events based on text description (Kilgour et al., 2022; Liu et al., 2022; 2023c; Li et al., 2023). These studies employ models like BERT (Devlin et al., 2019) (mini), CLAP, or other pre-trained models to comprehend descriptions of sound events, subsequently separating the sound sources consistent with the target description. However, they are not specifically designed for speech signals. In contrast to audio event classes, speech signals are considerably similar when observed from spectrograms, lacking clear acoustic spectral patterns to follow. Instead, they rely more on perceptual differences in auditory objects and semantic information. In addition to sound events, these models also focus on separating musical instruments (Chen et al., 2023; Huang et al., 2023a; Chen et al., 2023). It's important to note that while these previous works have made significant strides in the field, the specific challenges and nuances of speech signal separation present a unique problem space that our work aims to address. Labels, particularly those implemented via one-hot vectors, can be seen as a distinctive type of human language. In the realm of label-based audio/music/speech extraction systems (Manilow et al., 2020; Delcroix et al., 2021; Tzinis et al., 2022; Delcroix et al., 2023; Li et al., 2023; Ochiai et al., 2020), the works of Manilow et al. (2020) and Tzinis et al. (2022) are most closely aligned with ours. These systems, like ours, endeavor to integrate human subjective intentions into the separation process through attribute labels. Yet, they solely rely on one-hot vectors, resulting in a lack of flexibility within human-computer dialogue systems. In addition, they cannot understand the vast array of human language inputs and struggle significantly when dealing with open-ended queries. By contrast, we employ LLMs to understand cues that extend beyond human descriptions of auditory object differences, which offers increased flexibility in cue extraction. Furthermore, we've investigated control capabilities and made a connection between the perceptual differences of auditory objects in mixture and voiceprint systems. Another method utilizes semantic cues, such as images Ohishi et al. (2022), to extract speakers' speech discussing a particular concept. However, the necessity for corresponding images constrains its potential application domains.

## 6 CONCLUSION AND FUTURE WORKS

In this study, we proposed a novel paradigm for target speaker extraction, namely LLM-TSE, a significant departure from previous methodologies. Our approach uniquely introduces text to provide useful speaker extraction cues, which is an innovation that has demonstrated notable success and improvement in our experimental results. Our investigations have illuminated the potential of natural language to provide a rich source of discriminative features. These features can be leveraged independently as extraction cues, showcasing the versatility and effectiveness of natural language in this context. Furthermore, natural language is useful for performing task selection, which represents a promising approach to achieving auditory attention switching. Moreover, our paradigm augments the performance of audio-only systems by integrating contextual information from the present acoustic environment, which is often overlooked in traditional methods. This addition provides a more comprehensive and accurate representation of the target speaker's context, further enhancing the extraction process. In summary, our proposed paradigm signifies an important advancement for target speaker extraction systems, extending accessibility and improving performance. Not only does it provide a fresh perspective on the extraction process, but it also lays the groundwork for potential future studies on the cocktail party problem. Moving forward, we plan to persist in this direction, enhancing machines' ability to understand the foundations of human perception of multiple auditory objects within complex acoustic environments using natural language cues. Specifically, we aim to incorporate a range of mutually exclusive or non-exclusive auditory attributes, label flexible and open-ended text descriptions, and develop the capability for multi-round separation.

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

## A APPENDIX

### A.1 IMPLEMENTATION DETAILS

**Model Architecture:** The LLM-TSE model incorporates a text cue encoder derived from the LLaMA-2 7B model, a transformer decoder architecture. We generate the text cue embedding using the averaging results of the outputs of the last four self-attention layers. Subsequently, a linear

projection layer is employed to map its dimensions to match the embedding output of the audio cue encoder model. The construction of the audio cue encoder and extractor is built upon an open source of the time-domain SpeakerBeam (TD-SpeakerBeam) [5]. The default model hyperparameters from TD-SpeakerBeam are employed in this process.

**Optimization:** We use the AdamW optimizer for optimization, with an initial learning rate of 1e-4, which has proven effective for various tasks in our preliminary experiments. Our model is trained using ten NVIDIA 3090 GPUs, each with a batch size of 1. For stable training, we employ gradient accumulation, with backpropagation performed every two interactions, culminating in a valid batch size of 40 per iteration. A linear warmup scheduler is used for the first 1000 iteration steps, during which the learning increases from 0 to 1e-4 and remains constant. This strategy aims to gradually prepare the model for more complex tasks and improve overall learning stability. Finally, based on our preliminary experiments on the current dataset, we use the gradient normalization with a value of 30. This operation controls the weight update step and prevents gradient explosion.

**LoRA Adaptor:** We adopt the LoRA approach for efficient fine-tuning. The hyperparameters of the LoRA matrix, rank $r$, and scaling weight $\alpha$ are set to 16 and 16. The LoRA dropout is set to 0.05. These LoRA adaptors are applied to the linear projection layers of the query and key calculation in the self-attention layers.

**Generation of Prompts:** We will first write ten question sentences by hand for each task. Then, we used ChatGPT-4-32K to rephrase and generate more questions for each task. The prompt of rephrase is: "Keep it short, limit to 8 words. Feel free to vary sentence structures, but avoid duplications, and synonyms can be replaced. Imitate the tone of a casual conversation, don't be too rigid. Maintain the existing JSON format when outputting."

## A.2 LIMITATIONS

While our study introduces a paradigm for speaker extraction, it does not encompass all the acoustical characteristics and contextual cues that can be utilized for this purpose. Indeed, numerous features assist humans in distinguishing auditory objects that we have not fully explored in this research. For instance, pitch, timbre, speech speed rate, and rhythm can provide significant cues for differentiating between speakers (Haykin & Chen, 2005; Mesgarani & Chang, 2012; Popelka et al., 2016). High-level Semantic Information, such as the topic of conversation (e.g., "the person talking about the topic of weather"), can also serve as a powerful identifier. Furthermore, the speaker's age often influences the characteristics of their speech and can thus be a valuable cue for speaker extraction. These are all potent cues that could be further explored and integrated into our system. Future work could focus on extending the model's capability to handle these additional attributes, refining the methodology for incorporating such diverse information and evaluating the subsequent improvements in system performance. Moreover, while our proposed system shows promise, there are potential challenges in implementing and testing the system in real-world scenarios, such as in noisy environments or situations with multiple concurrent speakers. Further research is therefore needed to evaluate the system under such conditions and to develop strategies for dealing with these challenges.

In the future, we aim to delve into more open-ended perceptual concepts. Currently, our work is constrained by a reliance on predefined categories, and we cannot handle relatively abstract perceptual descriptions of auditory objects. For example, a description such as "The first speaker's voice is quite resonant, but after discussing basketball, the voice gradually diminishes" is beyond our current system's capacity. However, we believe that dealing with more open-ended and advanced problems necessitates a foundational understanding of the basic attributes of auditory objects. This understanding forms the core of our future work. We are committed to exploring how to integrate more open and detailed descriptions of auditory object differences into speech separation scenarios.

---

[5]https://github.com/BUTSpeechFIT/speakerbeam

