# OpenReview forum: "Typing to Listen at the Cocktail Party: Text-Guided Target Speaker Extraction"
_ICLR.cc/2024/Conference — Submitted to ICLR 2024_

### Official Review · Reviewer_ZMpA · 2023-10-28

**Soundness:** 2 fair
**Presentation:** 3 good
**Contribution:** 2 fair
**Rating:** 3
**Confidence:** 4

**Summary:**

This paper deals with the problem of target speaker extraction, where a speech by a speaker having the designated properties is extracted from speech mixtures. Although the previous methods for this task provide a cue of the speaker properties as speeches or images, the proposed method employs text prompts instead and encodes them with LLMs (more specifically, LLaMA-2). Once we can obtain feature embedding for representing the speaker properties, we can extract the target speech by injecting this feature embedding into the existing target speaker extractors. Experimental evaluations with the original dataset demonstrate that the proposed method outperformed the existing target speaker extraction named TD-SpeakerBeam in terms of SDR.

**Strengths:**

1. Target speaker (or speech) extraction is one of the hot topics in audio signal processing and many excellent techniques have already been proposed. This work extends those existing methods by introducing text prompts. As presented in Section 2, text prompts (1) are intuitive for humans, (2) have flexibility for designating the target speaker property, and (3) can make effective use of emerging large language models with amazing abilities.

2. This paper is well written, well structured and easy to follow.

**Weaknesses:**

1. I could not understand what is the most significant technical hurdles for incorporating text prompts into the existing target speaker extraction methods.
    - As far as I know, almost all the previous methods rely on feature embeddings for representing speaker properties to be extracted, which indicates that we do not mind which modality should be used for speaker representations. More specifically, universal sound selector [Ochiai+ Interspeech2022] has almost the same structure as the proposed model presented in Figure 3. This type of integration seems to be natural if we try to feed text prompts into target speaker extraction.
    - The proposed method provides a simple baseline for text-based target speaker extraction. However, I could not find any special tricks for this purpose.

2. Experimental comparisons with other methods should be presented.
    - I understand that the authors could not find any other baselines for text-based target speaker extraction. However, the author should introduce a simple baseline different from the proposed method for demonstrating the effectiveness of the proposed method (but it might be difficult since the proposed method itself is a naive baseline).
    - As presented before, universal target selector [Ochiai+ Interspeech2022] can be easily applied to text-based target speaker extraction. If the authors believe that the proposed method has a technical novelty against this work, experimental comparisons with it is mandatory for demonstrating the effectiveness.
    - Also, ConceptBeam [Ohishi+ ACMMM2022] presented a different network architecture for target speech extraction. Although this work mainly focuses on the use of images for representing the speech property, it can be easily applied to other modalities such as texts.

**Questions:**

Please check the above Weakness part.

---

> ### Author Response · Authors · 2023-11-20
> **Response to Reviewer ZMpA (1/2)**
>
> We appreciate your comments and would like to address your concerns with our following responses.
>
> ## Response to Weakness 1
>
> Thank the reviewer for these insightful comments. Since the inception of SpeakerBeam [1], target speaker extraction systems have adhered to the encoder + fusion + extractor framework. Many top-ranked approaches in the Microsoft Deep Noise Suppression Challenge at ICASSP 2023 adopt this structure. Therefore, this paper does not question or modify this established framework. Instead, we aim to address the inherent limitations of the entire target speaker extraction process and to chart new territories of possibility. Existing target speaker extraction (TSE) systems primarily face limitations related to privacy and dependency on high-quality voice prints. Specifically, the need for significant amounts of recorded voice raises privacy concerns. Additionally, the systems' performance can degrade significantly when suitably recorded voices are unavailable. In this work, we aim to address these difficulties in leveraging text-based cues for target speaker extraction by addressing the following two research questions:
>
> 1. What kind of text cues are useful for target speaker extraction? We devised text cues (absolute/relative/semantic content) inspired by human auditory scene analysis.
> 2. How do we enable the target speaker extraction system to understand these text cues? We attempted to use LLMs as a tool for text understanding and adopting LoRA-related fine-tuning techniques to achieve this.
>
> We believe notable works are typically simple and direct, yet they can potentially create substantial impact or yield transformative changes. Our research addresses many challenges and uncovers a wealth of exciting opportunities that are currently inadequately supported by existing target speaker extraction systems, as outlined below:
>
> - **Privacy issue**: Privacy has always been a public concern, especially in using a speaker's voice. However, voiceprints are critical identification information. Mainstream systems for speaker extraction, which predominantly rely on registered speech, would demand up to 10~30 seconds of recorded voice, e.g., a series of extraction systems, such as [2] and [3], are among the top-ranked extraction systems in Microsoft Deep Noise Suppression Challenge at ICASSP 2023. This requirement for users to provide registered speech when implementing mixed speaker extraction presents a significant challenge. It raises considerable concern, as individuals are naturally reticent about their voices being commodified and circulated in the marketplace. The landscape changes when we consider text information as an alternative. Unlike voiceprints, text does not carry personal identity information, making it a substantially more acceptable option. In addition, the text is undoubtedly the least costly compared to cues of other modalities, e.g., target angle, image, and video.
> - **Length of registered voices**: Our obstacles extend beyond privacy. The efficacy and applicability of the extraction system are also impeded by its dependency on registered voices. In numerous instances, obtaining high-quality and sufficiently lengthy registered voices proves to be a daunting task. We may find ourselves in situations where no registered voices are available. Such circumstances can lead to a substantial degradation in the performance of the target speaker extraction system [1] and complete system failure in the most extreme cases. In response to this challenge, our research ventured into uncharted territory. We recommend incorporating text as an independent extraction cue or in tandem with the registered voice. This represents a pioneering and inventive approach that has not been recognized or explored in the solution strategies.
> - **First to handle speech**: While text has been employed to extract audio events around or even before the development of this paper [4], the issue of speaker extraction has persistently lacked an effective resolution. This is a particularly critical concern as voice forms the foundational mode of human communication. Compared with audio events, which each possess unique and distinct spectral structures, the spectral structures of different voices are strikingly similar. This fundamental difference in our goals is akin to distinguishing between the barks of a Teddy Bear dog and a Samoyed, as opposed to simply differentiating between human voices and dog barks, as the sound extraction model might. To address this issue, we have proposed a novel method that pivots on metadata extraction, considering elements such as absolute or relative attributes and semantic content. This represents the first attempt to use text to navigate these complex challenges. The design of our methodology stems from an intrinsic understanding of how humans distinguish between different auditory objects.

---

> ### Author Response · Authors · 2023-11-20
> **Response to Reviewer ZMpA (2/2)**
>
> (Following the previous)
>
> - **Text as a valuable auxiliary tool**: Our experimental findings indicate that text can indeed function as a valuable auxiliary tool, bolstering the efficacy of target speaker extraction systems. It's crucial to acknowledge that registered voices typically require prior recording. In addition, the speech signal of the same speaker might have highly different characteristics in different conditions due to such factors as acoustic environment or emotional state. It is very challenging to make TSE systems robust enough for such intra-speaker variability [5]. Nevertheless, we can employ text to inform the model about the current state of the speaker within the mixture of speech. This method is undeniably more direct and efficient, providing us with a robust tool to tackle the challenge of voice extraction more effectively.
>
> ## Response to Weakness 2
>
> Thank you for suggesting the two relevant works. We have added them to the manuscript.
> We appreciate your feedback and acknowledge the challenges in identifying an appropriate baseline for comparison.
>
> The work [Ochiai] centers on audio event extraction, utilizing labels of each audio event as extraction cues. As previously stated, the foundational framework for target speaker/audio event extraction is well established. This is also grounded in the SpeakerBeam model. The innovation of this research primarily lies in its use of acoustic event classes to extract sound events. However, this model is not directly transferrable to speech. In sound events, all speakers are classified into a single category. In addition, this approach cannot focus on the content of the speech, such as through descriptions embedded in transcription snippets. It also does not accommodate general natural language queries. To establish a comparison with this approach, which employs one-hot representation, we removed data in our dataset associated with extraction through transcription snippets. We only kept the data can be distilled as a one-hot representation to demonstrate the discriminative capabilities of LLMs. The updated experimental results are shown here.
>
> | Entry   | Input (Audio) | Input (Text) | Transcription Snippet (50%, 80%, 100%) | Gender | Language | Far-near | Loudness |
> | ------- | ------------- | ------------ | -------------------------------------- | ------ | -------- | -------- | -------- |
> | Universal Sound Selector (Ochiai et al.)       | N             | One-Hot      | No Support                             | 10.54  | 8.88     | 10.25    | 8.96     |
> | LLM-TSE | Y             | N            | 7.30                                   | 10. 17 | 8.87     | 9.77     | 7.75     |
> | LLM-TSE        | N             | Y            | 2.70, 3.97,7.48                        | 10.40  | 9.38     | 10.57    | 8.89     |
> | LLM-TSE        | **Y**             | **Y**            | **7.96,9.81,10.05**                        | **10.87**  | **9.72**     | **10.66**    | **9.41**     |
>
> Noteworthy, however, using one-hot representations introduces the limitations:
> - One-hot representations can only express attributes with clear classifications, e.g., language/gender/loudness. However, if we want to use other cues, like transcription snippets, one-hot representations are unable to accomplish this
> - One-hot representations lack flexibility. Large language models (LLMs) can assist the target speaker extraction system understand user text inputs, allowing for more generic and diverse expressions. For example, the input of LLM-TSE can be easily extended to open-ended questions, such as "separating the speaker based on the 3-4 second segment in the mixed speech," something that one-hot representations are completely incapable of.
>
> We appreciate that you pointed out ConceptBeam, a system designed to operate in situations where multiple speakers are discussing various semantic concepts or topics within a mixed audio setting. The system leverages related images to the discussed topic to aid speech extraction. While the architecture of this system, which incorporates an encoder, a fusion module, and an extractor, does bear some similarity to our design, the operational conditions differ significantly. It's important to highlight that the functionality of ConceptBeam is contingent upon the availability of pair-matched images that correspond with the semantic content of the speech. This condition isn't typically satisfied in many practical applications. This unique prerequisite significantly constrains the adaptability of the ConceptBeam system when compared with methods that employ registered voices or our proposed text-based approach.
>
> [1] K. Žmolíková et al., SpeakerBeam, IEEE JSTSP
>
> [2] Y. Ju et al., TEA-PSE 3.0, ICASSP 2023
>
> [3] J. Yu et al., TSpeech-AI System, ICASSP 2023
>
> [4] C. Li et al., Target sound extraction, ICASSP 2023
>
> [5] Žmolíková et al, Neural Target Speech Extraction: An overview, IEEE SPM

---

> ### Comment · Reviewer_ZMpA · 2023-11-23
> **Thanks for the effort**
>
> Thanks for the effort in the rebuttal. I have read through all the author replies and I found that all the replies are reasonable.

---

> > ### Author Response · Authors · 2023-11-23
> > **A Gentle Reminder for Final Hours for Discussion**
> >
> > Dear reviewer ZMpA,
> >
> > Thank you for your time and effort in reviewing our paper. As we near the end of the discussion period, please let us know if you have any remaining questions or concerns. We have diligently addressed your previous comments, including results from new experiments.
> >
> > Feel free to reach out with any final feedback in the next few hours. Your insights are greatly valued. Thank you again for your thorough review and suggestions.
> >
> > Sincerely,
> >
> > Submission5734 Authors

---

### Official Review · Reviewer_tdLn · 2023-10-30

**Soundness:** 3 good
**Presentation:** 3 good
**Contribution:** 2 fair
**Rating:** 5
**Confidence:** 4

**Summary:**

This paper presents a target speech extraction (TSE) method that uses text-based cues from a large language model (LLM).
The model consists of a standard masking extractor with encoder and decoder, where the mask is generated using an audio cue embedding of enrollment speech as well as text cue from the LLM.
The model is trained with a range of text prompts to deal with various instructions for TSE tasks.

**Strengths:**

* The use of LLM enables the framework to handle flexible instructions and rich information to extract the target speech.
* The method demonstrates improved separation performance in terms of SI-SDR score.

**Weaknesses:**

The method is technically sound but exhibits a marginal novelty.
Mask generation using embeddings of auxiliary information is a common practice today such as (Liu et al. 2022; Oishi et al. 2023).

What has become fundamentally possible given the power of LLMs?
If we restrict some form of templates of prompts to feed into the LLM for a handful of separation scenarios to specify gender/language/loudness etc., such information may be provided as one-hot representations.

**Questions:**

The ablation study in Table 1 shows some variations of LLM-TSE: audio + text, text-only and audio-only.
1. Can we train the model with audio + text and switch to text-only or audio-only in inference time?
It seems the embedding vectors are concatenated and used for mask generation.
In this case, I am wondering how the lack of either modality is handled.

2. How does audio-only LLM-TSE differ from TD-SpeakerBeam?

---

> ### Author Response · Authors · 2023-11-20
> **Response to Reviewer tdLn**
>
> ## Response to Weakness
>
> We appreciate your insightful comments and the opportunity to clarify our work's scope. As our first venture into text-guided target speaker extraction, we aimed to showcase the potential of our proposed framework by examining a wide range of application scenarios and auditory cues. These include absolute attributes (like language and gender), relative attributes (such as loudness and distance), and semantic cues (i.e., transcription snippets). These cues are extensively used by the human auditory system to tackle the cocktail party problem [1].  Text-based systems open up new possibilities and address issues inherent in traditional target speaker extraction systems:
>
> 1. Resolving privacy concerns by using text instead of voiceprints.
> 2. Reducing dependency on the length of registered voices by using text as an extraction cue.
> 3. Use text can serve as a valuable auxiliary tool to improve the efficacy of target speaker extraction systems, overcoming intra-speaker variability by informing the model about the speaker's current state.
>
> While one-hot representations can handle some of these scenarios, they have notable limitations when dealing with text-based cues. Specifically, one-hot representations are constrained to express attributes with distinct classifications, such as language, gender, and loudness. However, they fall short when encoding more nuanced cues, such as transcription snippets that remain less disrupted in an audio sample.
>
> In contrast, Large Language Models (LLMs) can help the target speaker extraction system understand user text inputs, thereby enabling more generic and diverse text-based cues. For instance, under our proposed framework, the input of LLM-TSE can be easily extended to open-ended questions like "separate the speaker based on the 3-4 second segment in the mixed speech," a task beyond the capabilities of one-hot representations. Similarly, It has the potential to handle complex requests like "I want to extract the speaker who initially spoke loudly about basketball, but later his voice softened, and his speech speed slowed down as he moved further from the microphone." Thus, the proposed LLM-based target speaker extraction framework offers a more adaptable and nuanced approach to speaker extraction, addressing the limitations of existing methods. We believe that our work could have a substantial impact on the field.
>
> Following your question, we added this part (distilling the text prompts into one-hot representations) of the experiment to confirm the LLM's ability to understand the text prompts related to speaker extraction. The integration of the one-hot representation is referenced from the work of Universal Sound Selector (Ochiai et al.) [2]. The performance of the text-based system is comparable to that of the one-hot-based system, but it supports more tasks.
>
> | Entry   | Input (Audio) | Input (Text) | Transcription Snippet (50%, 80%, 100%) | Gender | Language | Far-near | Loudness |
> | ------- | ------------- | ------------ | -------------------------------------- | ------ | -------- | -------- | -------- |
> | Universal Sound Selector (Ochiai et al.) [2]       | N             | One-Hot      | No Support                             | 10.54  | 8.88     | 10.25    | 8.96     |
> | LLM-TSE | Y             | N            | 7.30                                   | 10. 17 | 8.87     | 9.77     | 7.75     |
> | LLM-TSE        | N             | Y            | 2.70, 3.97,7.48                        | 10.40  | 9.38     | 10.57    | 8.89     |
> | LLM-TSE        | **Y**             | **Y**            | **7.96,9.81,10.05**                        | **10.87**  | **9.72**     | **10.66**    | **9.41**     |
>
>
> ## Response to Question 1
>
> Thank you for your question. Our model can handle the scenarios where either modality is missing. Specifically, our proposed LLM-TSE model uses a combination of audio+text (50%), text-only (20%), and audio-only (30%) data for training. After the model had been trained, we tested it across three scenarios: audio-only, text-only, and audio+text, as shown in Table 1.  This presents a unique advantage of our system, as the quality of the registered audio might be poor or missing due to privacy concerns. In these cases, the proposed LLM-TSE model can provide additional text-based cues to augment the performance of target speaker extraction.
>
> ## Response to Question 2
>
> For a fair comparison, our audio-only LLM-TSE is the same as TD-SpeakerBeam, and we have its open-sourced code from https://github.com/BUTSpeechFIT/speakerbeam. We would like to highlight that instead of improving the audio TSE model, our main objective in this work lies in incorporating text-based cues to address the issues of the target speaker extraction system.
>
> [1] The What, Where and How of Auditory-object Perception, Bizley et al., Nature Reviews Neuroscience.
>
> [2] Listen to What You Want: Neural Network-based Universal Sound Selector, Ochiai et al. Interspeech 2020.

---

> > ### Author Response · Authors · 2023-11-23
> > **A Gentle Reminder for Final Hours for Discussion**
> >
> > Dear reviewer tdLn,
> >
> > We deeply appreciate the time and effort you have invested in evaluating our manuscript. As we approach the close of the discussion phase, we would be grateful if you could inform us about any additional questions you may have. We have earnestly taken your prior remarks into account and have incorporated findings from new experiments accordingly.
> >
> > Please do not hesitate to provide any concluding comments within the forthcoming hours. Your valuable input is highly esteemed. Once again, we express our gratitude for your meticulous review and constructive feedback.
> >
> > Sincerely,
> >
> > Submission5734 Authors

---

### Official Review · Reviewer_K7zB · 2023-11-07

**Soundness:** 3 good
**Presentation:** 3 good
**Contribution:** 3 good
**Rating:** 8
**Confidence:** 5

**Summary:**

The paper proposes a new approach for target speaker extraction called LLM-TSE that incorporates natural language input to guide the extraction process. This aims to enhance flexibility and performance compared to existing methods that rely solely on pre-registered voiceprints.

The user can provide text input describing various cues about the target speaker, such as gender, language, volume, distance, or even transcription snippets. This text input is encoded by a large language model to extract useful semantic information.

The text embeddings are fused with optional pre-registered voiceprint embeddings and passed to an extractor module to selectively extract the target speaker from the input mixture.

Experiments demonstrate competitive performance using text cues alone, and SOTA results when combined with voiceprints. The method shows particular gains when text provides complementary contextual cues beyond the voiceprint.

**Strengths:**

Enhanced flexibility: Can utilize text cues alone without needing pre-registered voiceprints. Allows incorporating a wide range of perceptual cues through natural language descriptions.

Improved controllability: Text input can be used to direct the model to extract or remove a target speaker, going beyond just extracting a pre-registered voice.

SOTA performance: Achieves top results on benchmark datasets, outperforming previous target speaker extraction methods.
Robustness to acoustic mismatches - Integrating contextual cues from text descriptions enhances robustness when enrollment conditions differ from test conditions.

Broadened applicability: Relies less on requiring voiceprints a priori, expanding applicability to more real-world scenarios where pre-registration is unavailable.

Novel paradigm: This signifies an important advancement in guided and adaptable target speaker extraction, laying the groundwork for future cocktail party research.

Leverages large language model: Utilizes powerful pre-trained LLM to effectively interpret semantic concepts from natural language descriptions.

**Weaknesses:**

Lack of psychoacoustic analysis: No analysis related to human auditory perception. For example, see the ICASSP 2023 Deep Noise Suppression challenge.

Limited perceptual cues: Does not yet handle more complex cues like pitch, emotion, timbre, age, topic of conversation, etc. Relies on predefined attributes.

Evaluation on simulated data: Performance needs further evaluation on real-world noisy conditions with multiple concurrent speakers. A real-world test set like used in the ICASSP 2023 Deep Noise Suppression Challenge should be used.

Results (Table 1) are compared with only two other models. It is a stretch to say this is SOTA.

Constrained to simple descriptions: Cannot handle abstract or open-ended perceptual descriptions beyond basic attributes.

Computational complexity: Large language models have high computational costs.

Brittleness of LLMs: LLMs can exhibit biased and unreliable behavior. Robustness needs verification.

Single speaker extraction: Framework focused on extracting one target speaker, not multiple.

**Questions:**

None

---

> ### Author Response · Authors · 2023-11-21
> **Response to Reviewer K7zB**
>
> Thank you for your encouraging review and insightful suggestions.
>
> We have carefully considered each of your comments and are grateful for the expertise you have brought to this evaluation. We believe they provide valuable directions for future research. We plan to delve deeply into these areas in our continued work to make meaningful contributions to the field. For clarity and completeness, we have added a section in the revised manuscript (**Section A.2**) discussing these future directions. We believe this not only acknowledges the potential of your suggestions but also provides a roadmap for researchers who may wish to build upon our work.
>
> Once again, we are grateful for your time, expertise, and constructive critique that will undoubtedly help improve our research in the future.

---

> > ### Author Response · Authors · 2023-11-23
> >
> > Dear reviewer K7zB,
> >
> > Our team offers sincere thanks for the considerable time and effort you have dedicated to scrutinizing our research paper. As we close to the end of our discussion period, we kindly request you to share any further questions or issues. We extend our thanks once more for your comprehensive review and valuable suggestions.
> >
> > Warm regards,
> >
> > Submission5734 Authors

---

### Official Review · Reviewer_gyG3 · 2023-11-09

**Soundness:** 2 fair
**Presentation:** 3 good
**Contribution:** 3 good
**Rating:** 6
**Confidence:** 4

**Summary:**

This paper proposed to tackle the target speaker extraction problem by using text-guided approach. Compared to the conventional target speaker extraction that uses the enrolled speech from a specific speaker, the proposed text-guided approach aimed at instructing the model with input text information. A large language model is used to extract semantic cues from text information which is either fused with the audio cues or acting interpedently when used as the prompt for extracting the target speaker’s voice. These input texts were generated by creating some question templates and then expanding it using ChagGPT-3.5-Turbo. The experiment part demonstrates better SI-SDR performance compared to a baseline system.

**Strengths:**

1, The paper is well written, and the presentation is clear.
2, The method is innovative which enables the interaction of text prompt and the target speaker extraction.

**Weaknesses:**

1, Lack of references and comparison methods. Text-guided speech extraction is essentially a multi-modal speech extraction method, including text and speech modalities. The motivation is to leverage multiple distinctive clues to extract the target speech sound. While not using exact text and speech, such multi-modality-based speech extraction paper has been proposed before [1][2] which include text, image, and sound modalities. It’s necessary to compare such existing methods in terms of performance rather than only comparing it with the speech-only-guided target speech extraction method.

2, The experiment design is limited in the sense that these text prompt such as gender, language, far-near and loudness, are not typical cues which can be widely used for target speaker extraction. For example, if all the speakers are of the same gender in a conversation, which target speaker will be extracted? If all speakers speak the same language, how to extract the target speaker? For these scenarios, speakers are of different genders or speaking different languages, the target speaker extraction problem is indeed an easy problem. A more practical design scheme is needed for the input text.

ref
[1] Ohishi, “ConceptBeam: concept driven target speech extraction”, ACMMM, 2022.
[2] Li et al., “Target sound extraction with variable cross-modality cues”, ICASSP, 2023.

**Questions:**

For the audio cue encoder, have you tried a pre-trained speaker embedding module, such as d-vector model used for speaker recognition?

---

> ### Author Response · Authors · 2023-11-20
> **Response to Reviewer gyG3 (1/2)**
>
> We appreciate your comments and would like to address your concerns with our following responses.
>
> ## Response to Weakness 1
>
> Thank you for suggesting the two relevant works. We have added them to the manuscript. We fully agree on the value of comparative evaluations. However, it's important to note that our method and those introduced by [Ohishi et al.] and [Li et al.] are designed to **address significantly different tasks**. Consequently, a direct comparison might not provide a meaningful or equitable evaluation of their merits and capabilities.
>
> ###  Differences From the Work of [Li et al.]
> First of all, as you have mentioned, [Li et al.] have made strides in audio event extraction. However, our research focus diverges significantly from theirs. [Li et al.]'s model, while effective in its domain, does not accommodate the complexities inherent in speaker extraction, which is the main objective of our work. Their model treats speech as a singular, undifferentiated category. In contrast, our model is specifically engineered to identify individual speakers within the broad realm of speech. This fundamental difference in our goals is akin to distinguishing between the barks of a Teddy Bear dog and a Samoyed, as opposed to simply differentiating between human voices and dog barks, as [Li et al.]'s model might. This illustrates the intricate challenges our model is designed to overcome.
>
> Our work specifically addresses unique complications introduced by speech extraction, such as:
>
> 1. Determining effective text prompts for speaker extraction.
> 2. Ensuring comprehension of text inputs, which can be highly variable.
> 3. Navigating the interaction between text and speech cues. Should our model work with speech cues, as is common, or could it function solely on text, a route necessitated by difficulties in acquiring high-quality, registered voices or privacy concerns?
>
> Our model is innovative in its application of text as a tool to tackle these challenges, drawing on human perception of auditory differences. This presents a substantial technical hurdle. Future work could explore additional cues like open-ended questions to enhance our system's performance further. We believe this provides a clearer picture of the distinctiveness and significance of our work.
>
> ### Differences From ConceptBeam
>
> ConceptBeam is indeed a system designed to operate in situations where multiple speakers are discussing various semantic concepts or topics within a mixed audio setting. The system leverages related images to the discussed topic to aid speech extraction. While the architecture of this system, which incorporates an encoder, a fusion module, and an extractor, does bear some similarity to our design, the operational conditions differ significantly. It's important to highlight that the functionality of ConceptBeam is contingent upon the availability of pair-matched images that correspond with the semantic content of the speech. This condition isn't typically satisfied in many practical applications. This unique prerequisite significantly constrains the adaptability of the ConceptBeam system when compared with methods that employ registered voices or our proposed text-based approach.
>
> ### Compare with One-Hot System
>
> While a direct comparison with these methods may not be feasible, we drew inspiration from [Li et al.]'s work and related Universal Sound Selector (Ochiai et al.) [3]  to encapsulate attribute-based questions (such as language, gender, loudness, and distance) into a one-hot representation. This was used as a baseline to assess the comprehension capabilities of Language Models (LLMs). However, we must acknowledge the limitations inherent in using one-hot representations:
> - One-hot representations are only capable of expressing attributes with distinct classifications, for instance, language, gender, and loudness. If we want to employ other cues, like transcription snippets, one-hot representations prove insufficient.
> - One-hot representations lack adaptability. LLMs can aid the target speaker extraction system in interpreting user text inputs, thus facilitating the injection of more generic and diverse semantic cues. For example, the input of LLM-TSE can be effortlessly extended to support open-ended questions, such as "isolate the speaker based on the 3-4 second segment in the mixed speech," a task beyond the capacity of one-hot representations.

---

> ### Author Response · Authors · 2023-11-20
> **Response to Reviewer gyG3 (2/2)**
>
> | Entry   | Input (Audio) | Input (Text) | Transcription Snippet (50%, 80%, 100%) | Gender | Language | Far-near | Loudness |
> | ------- | ------------- | ------------ | -------------------------------------- | ------ | -------- | -------- | -------- |
> | Universal Sound Selector (Ochiai et al.) [3]       | N             | One-Hot      | No Support                             | 10.54  | 8.88     | 10.25    | 8.96     |
> | LLM-TSE | Y             | N            | 7.30                                   | 10. 17 | 8.87     | 9.77     | 7.75     |
> | LLM-TSE        | N             | Y            | 2.70, 3.97,7.48                        | 10.40  | 9.38     | 10.57    | 8.89     |
> | LLM-TSE        | **Y**             | **Y**            | **7.96,9.81,10.05**                        | **10.87**  | **9.72**     | **10.66**    | **9.41**     |
>
> ## Response to Weakness 2
>
> Thank you for your insightful question, which allows us to clarify further the scope and potential of our proposed framework for real-world applications. As an initial venture into text-guided target speaker extraction, we aimed for a broad examination of application scenarios and auditory cues. Specifically, we scrutinized absolute attributes (like language and gender), relative attributes (such as loudness and distance), and semantic content (via transcription snippets). These are all extensively utilized by the human auditory system to tackle the cocktail party problem [4, 5].
>
> In real-world scenarios, when one type of auditory cue is insufficient to discern a sound source, humans naturally shift their focus to other cues. Our LLM-TSE model is motivated by this adaptive behavior. It is designed as a versatile text-based speaker extraction framework, capable of flexibly choosing and smoothly transitioning between auxiliary cues based on different situations.
>
> However, we agree with your assertion that refining the text prompts to cover more practical scenarios carries great significance. This indeed calls for substantial research efforts to design comprehensive evaluation benchmarks. As stated in our conclusion, we are actively integrating additional cues such as timbre, emotional state, sentiment scoring, speech velocity (calculated using Whisper timestamps), pitch, and energy (evaluated with Librosa). These improvements aim to enhance the applicability of our framework further. We look forward to sharing our progress in future work.
>
> ## Response to Questions
>
> Thank you for your valuable question. We have not yet experimented with pre-trained speaker embedding modules, such as d-vector or x-vector, as our audio cue encoder. While neural network-based embeddings like x-vectors or d-vectors are designed and trained specifically for speaker classification tasks and indeed contain speaker-specific information, it remains uncertain whether these representations are optimal for TSE tasks [6]. In our work, we've chosen a more common setting. Our audio cue encoder, which performs speaker embedding extraction, takes an enrollment utterance as input. It typically includes a pooling layer that converts frame-level features into a single vector, mirroring the functionality of the aforementioned embedding extractors. This neural network is trained concurrently with the main network using a shared objective function. The benefit of this approach is that the embeddings are trained explicitly for TSE, thus gathering crucial information specifically for this task. Contrastingly, pre-trained embedding extractors like d/x-vectors are often trained on larger corpora, potentially offering greater robustness. A potential compromise might involve utilizing a pre-trained embedding extractor and fine-tuning it with the TSE task. To our knowledge, this approach has not been explored yet. It would be interesting to explore these pre-trained speaker embeddings, which can potentially offer greater robustness as they have been trained on larger corpora. Due to the time limitation during the rebuttal, we will leave this study as a future work.
>
> [3] Listen to What You Want: Neural Network-based Universal Sound Selector, Ochiai et al. Interspeech 2020.
>
> [4] The What, Where and How of Auditory-object Perception, Bizley et al., Nature Reviews Neuroscience.
>
> [5] Selective Attention in Normal and Impaired Hearing, Shinn-Cunningham et al., Trends in Amplification.
>
> [6] Neural Target Speech Extraction: An overview, Zmolikova et al, IEEE Signal Processing Magazine.

---

> > ### Author Response · Authors · 2023-11-23
> >
> > Dear reviewer gyG3,
> >
> > We are sincerely grateful for the invaluable time and work you've committed to assessing our work. As we draw towards the end of this discussion phase, we hope you to share any further questions or issues you might still have. We've responded to your earlier comments diligently and have included the results from additional experiments.
> >
> > We would really appreciate that if you communicate any last-minute thoughts or feedback in the immediate hours. Your insightful perspectives are of great importance to us. We extend our thanks once more for your comprehensive review and valuable suggestions.
> >
> > With appreciation,
> >
> > Submission5734 Authors

---

### Author Response · Authors · 2023-11-23
**Summarize Contributions of the Paper**

Dear Associate Chair and Reviewers,

We would like to express our sincere gratitude for your dedicated time and effort in reviewing our paper. Your constructive suggestions have been instrumental in enhancing the quality of our manuscript. All reviewers endorse the effectiveness of the proposed mechanism. The reviewers gyG3, K7zB, and ZMpA have well recognized the paper presentation and writing.

We want to highlight the main contributions of our paper, including the additional insights gained during the rebuttal phase:

- Our paper presents the **first-ever approach** to introducing text-based target speaker extraction (TSE), effectively enhancing the feasibility, controllability, and performance of current TSE models. As indicated by our experimental results:
   - Text proves its capability to act as a standalone extraction cue, potentially addressing the **privacy** and **instability** issues inherent in predominant voiceprint-based target speaker extraction systems.
   - The use of text input allows the model to either extract or eliminate a target speaker, overcoming the constraints associated with extracting only pre-registered voices.
   - By informing TSE models about the speaker's current state, text can help tackle **intra-speaker variability**, thereby enhancing the effectiveness of speaker extraction.
- (Quoted from **reviewer K7zB**) Achieves **top results on benchmark datasets**, outperforming previous target speaker extraction methods. Robustness to acoustic mismatches - Integrating contextual cues from text descriptions enhances robustness when enrollment conditions differ from test conditions.
- (Quoted from **reviewer K7zB**) This **signifies an important advancement** in guided and adaptable target speaker extraction, laying the groundwork for future cocktail party research.

Following your feedback, we have further improved our manuscript and appendix. Again, we extend our heartfelt thanks for your valuable feedback and time.

With appreciation,

Submission5734 Authors

---

### Meta-Review · Area_Chair_Sp73 · 2023-12-05

**Metareview:**

The paper presents an approach for target speaker extraction that leverages natural language input to guide the process. This text input is encoded by a large language model to extract pertinent semantic information. The experiments show competitive performance when using text cues alone and SOTA results when combined with pre-registered acoustic cues. However, reviewers have raised concerns about the comparison with other methods, the limited innovation in the technologies used, and the absence of more perceptual cues like pitch and emotion in the text input.

**Justification For Why Not Higher Score:**

Reviewers have raised concerns about the comparison with other methods, the limited innovation in the technologies used, and the absence of more perceptual cues like pitch and emotion in the text input.

**Justification For Why Not Lower Score:**

N/A

---

### Decision · Program_Chairs · 2024-01-16

Reject